# Molecular Dynamics Simulation and Structure Changes of Polyester in Water and Non-Aqueous Solvents

**DOI:** 10.3390/ma15062148

**Published:** 2022-03-15

**Authors:** Jin Zheng, Dongshuang Wang, Qi Zhang, Meng Song, Mingli Jiao, Zhicheng Zhang

**Affiliations:** 1College of Textile, Zhongyuan Universtity of Technology, Zhengzhou 450007, China; 13303773269@163.com (D.W.); 15515886126@163.com (Q.Z.); chengzz369@163.com (Z.Z.); 2Textile and Clothing Collaborative Innovation Center of Henan Province, Zhengzhou 450007, China; 3Textile and Garment Industry Research Institute, Zhongyuan University of Technology, Zhengzhou 450007, China; 4School of Materials and Chemical Engineering, Zhongyuan University of Technology, Zhengzhou 450007, China; chengsimengyin@126.com (M.S.); johnml@163.com (M.J.)

**Keywords:** polyester, non-aqueous solvent, free volume, molecular dynamics

## Abstract

Studying the changes in the microstructure of polyester (PET) in water and non-aqueous solvents is important to understand the swelling mechanism of PET, which can help to reduce water pollution during the dyeing process. This study uses molecular models of PET, water, and decamethyl-cyclopentasiloxane (D5) and employs molecular dynamics method to simulate the influence of solvents on the microstructure of PET. The results show that the glass transition temperature (T_g_) of the pure PET system is close to the experimental value. The T_g_ of PET decreases with the addition of water and D5 solvents, and the free volume after adding D5 is higher compared to the free volume after adding water. Through molecular dynamics simulation of PET microstructure, it is found that D5 has a better SWELLING effect on PET than water.

## 1. Introduction

Polyethylene terephthalate (PET, polyester) fiber has a very tight structure and strong hydrophobicity. The cross-section of the PET fiber is circular under the microscope, its longitudinal thickness is uniform, and its surface is smooth [1,2,3]. During the PET dyeing process, the dispersed dye must enter the PET fiber through a narrow gap, which poses great difficulty. Therefore, to achieve the dyeing of PET [4,5,6], dyeing conditions of carrier, high temperature, or hot melt are generally required [7,8,9,10,11,12]. This kind of dyeing technology requires more equipment and higher cost. Moreover, the dyeing of polyester fabrics consumes a large amount of water, and the cost of waste treatment is high.

In order to solve the above problems, it is necessary to use other media with different properties from water. Hu [13] used 15 kinds of pure disperse red dyes to dye polyester/nylon microfiber fabrics for 60 min using self-developed supercritical dyeing equipment. Infrared spectral analysis showed that no new groups were introduced after the fabric was dyed, and the original properties of the fabric were maintained. Li et al. [14] reported that the non-aqueous solvent D5 has a certain swelling effect on PET. Wang et al. [15] added an accelerator to the D5 solvent dyeing system, and found that the accelerator increased the dye uptake rate of disperse dyes on PET. After measuring the fiber diameter with a confocal laser scanning microscope, they found that the accelerant improved the swelling of PET in silicone solvents. Therefore, it is necessary to study the influence of aqueous solvents and non-aqueous solvents on the microstructure of PET. The change in the microstructure of PET fiber is a necessary link in the dyeing process. To realize the dyeing of PET in solvent, the key lies in grasping the changes in PET microstructure in solvent. Research from a microscopic point of view to explore the variation of PET fiber structure under the action of water and non-aqueous solvents is conducive to deepening the knowledge and understanding of the dyeing mechanism.

Molecular dynamics (MD) simulation helps to understand the microstructure of polymers, analyze the changes in polymer structure at the molecular level, and provide new research methods [16,17]. MD simulation can explore the microscopic behavior of particles in the diffusion-limited aggregation process at the atomic level to reveal the aggregation mechanism of particles in aqueous solution [18]. MD simulation is also widely used to study the mechanism of interaction between different materials [19]. Sheng [20] studied the glass transition temperature of polyethylene/graphene nanocomposites through MD. The specific volumes of the three systems (polyethylene, polyethylene with small graphene flakes, and two small graphene flakes) were obtained as a function of temperature. It was found that the glass transition temperature decreases with the increase in content of graphene. Van der Waals energy changes significantly with the increase in content of graphene, and torsional energy also plays an important role in the glass transition of polymers. These results indicate that graphene can promote the movement of polymer segments and reduce the glass transition temperature of the polymer. Zhang et al. [21] used MD to study the swelling performance of polyvinyl alcohol (PVA) in water/ethanol solution, and found that pores containing water and ethanol were formed between the PVA chains. As the swelling degree increases, the number and size of the pores both increase. The diffusion coefficients of water and ethanol in swollen PVA increase linearly with the degree of swelling. Shanks and Pavel [22,23,24,25] used MD simulation to study the diffusion of methane and other gases and small molecule penetrants in aromatic PET, amorphous PET, and isomeric PET. They found that the diffusion coefficient and free volume and its distribution are related. Qiang et al. [26] used MD to study the cycling behavior of polyethylene (PE) under different loading conditions below the glass transition temperature at the nanoscale. The results show that van der Waals energy and dihedral angle energy are the primary factors affecting the cyclic behavior of PE. Wang et al. [27] simulated the glass transition behavior and mechanical properties of PET, PET/silica nanocomposites, and PET/hydroxylated silica nanocomposites. It was found that the content of silica had a certain influence on the T_g_ of PET. Moreover, Guo et al. [28] simulated the adsorption of sulfamethazine (SMT) on six microplastics, such as PET by MD simulation. Their results showed that the main mechanism of adsorption was electrostatic and Van der Waals interaction.

In this study, MD was used to study the changes in the microstructure of PET in the three systems of pure PET, PET/H_2_O, and PET/D5 at different temperatures. The swelling mechanism of PET was studied from a microscopic point of view, which can provide theoretical support for further research on PET dyeing with non-aqueous solvents.

## 2. Materials and Methods

The MD simulation was carried out using the Forcite and Amorphous Cell modules of Material Studio 8.0 software (Accelrys, San Diego, CA, USA) with the condensed-phase optimized molecular potentials for atomistic simulation studies (COMPASS, Center for Molecslar Science, Institute of Chemistry, Beijing, China) [29] force field, which is widely adopted to predict the structure of polymers. In the COMPASS force field, the total energy of the system is expressed as:(1)$$E_{\mathrm{total}}=E_{b}+E_{\theta }+E_{\varphi }+E_{\chi }+E_{\mathrm{cross}}+E_{\mathrm{ele}}+E_{\mathrm{vdw}}$$
where E_total_ is the total energy, E_b_ is bond stretching energy, and E_θ_ is angle bending energy. E_φ_ is dihedral torsion energy, E_χ_ is out-of-plane energy, E_cross_ is cross term interaction energy, E_ele_ is electrostatic interaction energy, and E_vdw_ is van der Waals interaction energy.

The electrostatic and van der Waals forces were calculated by using the atom-based summation method with a cut off value of 12.5 Å.

Figure 1 shows the construction process of the molecular simulation. First, a PET molecular chain containing 50 repeating units was constructed (Figure 1c), and the cell energy of each cell was minimized using the Smart Minimizer method. After the energy of the periodic cells was reduced through the geometric optimization process to obtain a more stable structure, the NPT (constant molecule, pressure, and temperature) ensemble was selected for annealing. The annealing process was performed for 20 cycles, and a total of 600 ps was run. After annealing, the system equilibrium was completed. First, 200 ps NPT ensemble simulation was conducted and then 500 ps NVT (constant molecule number, volume and temperature) ensemble simulation was carried out. Andersen [30] thermostat and Berendsen [31] barostat were used for temperature control and pressure control, respectively. System 2 and System 3 were subjected to the same simulation process. Finally, equilibrated simulation boxes were used to analyze the fraction of free volume (FFV) and other characteristic parameters (Figure 1g).

The basic parameters of PET and the combined system with H_2_O and D5 solvents are listed in Table 1. The volume and density of the polymer after optimization are also shown. The final density of the ensemble simulated PET system is 1.241 × 10^3^ kg/m^3^, which is close to the experimentally measured density of amorphous PET polymer (1.330 × 10^3^ kg/m^3^).

## 3. Results and Discussion

### 3.1. Evaluation of System Balance

In order to obtain reasonable parameters of the system, the system must be simulated for a long enough time for it to reach an equilibrium state. There are two main criteria for judging whether the system reaches equilibrium: one is temperature equilibrium, which requires that the fluctuation of the equilibrium value should not exceed 10 K. The second is energy balance, which requires the energy to be constant or fluctuate up and down along a constant value. Figure 2 presents the temperature and energy balance diagrams of PET under the NPT ensemble of 360 K and 150 ps, respectively. The standard deviation of the calculated temperature is 4.99 K. The standard deviations of the total energy, potential energy, non-bonding energy, and kinetic energy are 107.63 kcal/mol, 69.92 kcal/mol, 28.65 kcal/mol, and 65.51 kcal/mol, respectively, indicating that the energy of the simulated system has reached an equilibrium state.

### 3.2. Simulation Analysis of Glass Transition Temperature of PET

In this study, the glass transition temperature (T_g_) of Systems 1–3 was simulated and measured. MD simulation was used to determine the change curve of specific volume with temperature under constant pressure in the range of 200–500 K.

The initial stage temperature was set to 500 K, and the temperature was reduced by 20 K after each stage of the experiment was completed. The kinetic final equilibrium conformation of the previous stage was used as the initial conformation of the later stage MD simulation. At each temperature point, a 50 ps NVT ensemble dynamics simulation was performed, and then a 150 ps NPT ensemble dynamics simulation was performed. After the system reached equilibrium, data acquisition and analysis were performed.

The simulated specific volume versus temperature plot of PET is shown in Figure 3. The intersection of the two straight lines obtained by fitting is the glass transition temperature of the system [20]. Figure 3a shows that the glass transition temperature of PET is 370 K, which is close to the corresponding experimentally measured value of 342–350 K [32]. This indicates that the PET model established in this study can reflect the properties of the polymer, which further verifies its rationality for experiments. It can be seen from Figure 3b that the T_g_ of PET, after adding H_2_O solvent, is 368 K. Compared with the pure PET system, the temperature drop is smaller, indicating that the effect of water on reducing the T_g_ of PET is not obvious. It can be seen from Figure 3c that the T_g_ of PET and D5 system is 346 K. The simulation results show that the T_g_ of PET decreases significantly after adding D5, and the decrease of T_g_ is beneficial to reduce the dyeing temperature of PET and accelerate the dyeing of PET. This result indicates that the PET polymer model established in this paper can reflect the characteristics of the polymer, and the parameters used are correct, which is of instructive significance for conducting the related experiments. This also verified the rationality of the proposed model. On this basis, 413 K [33] was determined to be the suitable kinetic simulation temperature for analyzing the microstructure of PET in the dyeing process.

### 3.3. Simulation Analysis of Molecular Ability and Diffusion Characteristics of Polyester Molecular Chains

The mean square displacement (MSD) represents the mobility of particles in the system and is used to characterize the mobility of molecules. The calculation formula of MSD is as follows:(2)$$\mathrm{MSD}=\langle {\left|r_{i}\left(t\right)-r_{i}\left(0\right)\right|}^{2}\rangle$$
where r_i_(0) and r_i_(t) are the positions of particle i at the initial time and time t, respectively; the symbol 〈〉 indicates the average value for all particles.

The particle diffusion coefficient D can be calculated from the slope of MSD using the Einstein relationship as follows:(3)$$D=\underset{t\to \infty }{\lim}\frac{\langle {\left|r_{i}\left(t\right)-r_{i}\left(0\right)\right|}^{2}\rangle }{6t}$$

Figure 4 shows the MSD results of PET molecular chain in each of the three systems at a temperature of 413 K. It can be seen that the chain movement of the swollen PET macromolecule is greater than that of the PET molecular chain. In order to analyze molecular diffusion behavior more clearly, the MSD curves of three systems are calculated by Log10 and fitted by segments (as shown in Figure 5). In Figure 5, a is the slope of the fitting straight line, which is anomalous (non-Einstein) behavior when a < 1, and Einstein behavior when m is close to 1 [34]. As can be seen from the figure, at the back end of the simulation time, three systems transform to Einstein diffusion. During this time, the diffusion coefficient can be calculated by Einstein Equation (3). The diffusion coefficients of the three systems are shown in Table 2.

It can be seen from Table 2 that as the temperature increases, the molecular chain obtains more energy, so the movement is more active. At the same time, the higher the temperature, the softer the PET molecular chain. Consequently, more free volume is generated, and more space or path is provided for the diffusion of solvent molecules. In water and D5, the diffusion coefficient of D5 is higher than that of water.

### 3.4. Simulation Analysis of PET Microvoids and Their Distribution

Free volume (FV) refers to the volume in the system that is not occupied by molecules. It is an important parameter used to characterize the structure of polymers. The molecular chains can only move due to the existence of free volume. FFV is the percentage of free volume to the total volume of the system.

Figure 6 shows the schematic of van der Waals surface and Connolly surface. The van der Waals surface intersects with the van der Waals radii of the atoms in the structure and Connolly surface is at the boundary between the Connolly probe and the atoms. Free volume is defined as the volume circled by the Connolly surface. FFV was calculated by the ratio of free volume to the total volume of the model. In this study, the Atom Volumes and Surfaces tool in MS software was used to obtain the accessible FV of the probe by the Connolly surface method. The free volume was calculated when the probe radius was r and r + Δr in turn. The difference between the two is the free volume of the cavity with radius r ~ r + Δr.
(4)FFV(Δr)=FFV(r)−FFV(r+Δr) 

In the above formula, FFV(r) and FFV(r + △r), respectively, represent the percentage of the free volume that can be contacted when the probe radius is r and r + △r to the total volume of the system; FFV(△r) represents the free volume fraction of holes in the range of r ~ r + Δr. Figure 7 shows the volume morphology with different diameters at the same temperature. The blue and white areas represent the free volume created by transient gaps caused by inefficient chain packing or chain rearrangement. The free volume decreases with increase in probe diameter.

Figure 8 shows the free volume fraction of PET in the three systems at different temperatures. It can be seen that, as the temperature rises over the range from 400 K to 440 K, the free volume fraction of PET increases. In this temperature range, the free volume fraction of PET increases after adding water, indicating that the addition of water has a certain swelling effect on PET. However, the increase in free volume after adding D5 is larger.

In order to further explore the influence of H_2_O and D5 on the change in free volume of PET, the distribution of cavities in PET was studied. Figure 8a shows the distribution of free volume fractions in the three systems at a temperature of 413 K. The total free volume fractions FFV of PET are 37.459%, 37.560%, and 37.892%, respectively, indicating that the presence of water and non-aqueous solvent D5 increases the free volume. The presence of D5 significantly increases the total free volume of PET. It can also be seen from Figure 8b that the major changes in the free volume fractions of the three systems are mainly due to the cavities in the range of 1.0 Å to 4.0 Å in diameter. The contribution of the increase in the total free volume fraction is also mainly from the cavities in the range of 1.0 Å to 4.0 Å. The MD simulation results show that the non-aqueous solvent D5 has a better swelling effect on PET than water.

### 3.5. Interfacial Interaction Model and MD Simulation

The binding energy (E_Binding_) is an effective parameter to quantitatively characterize the strength of interactions between different substances. The total energy (E_total_) of the simulation system includes non-bond interaction energy (E_non-bond_), valence energy (E_valence_), and valence energy (cross terms) (E_crossterm_). The E_non-bond_ is the most important part of the total energy of the simulation system. The E_non-bond_ is composed of three important parts, including Van der Waals energy, electrostatic energy, and hydrogen bonding energy. However, the hydrogen bond energy calculated by the COMPASS force field is not listed separately, and it is included with other non-bond interaction terms [35]. The E_total_ of the simulation system can be expressed by the following Equation (5):(5)$$E_{\mathrm{total}}{=E}_{\mathrm{non}-\mathrm{bond}}{+E}_{\mathrm{valence}}{+E}_{\mathrm{crossterm}}$$

The binding energy between PET and H_2_O can be expressed by the following Equation (6) [36]:(6)$$E_{\mathrm{binding}}=-E_{\mathrm{Interaction}}=-\left(E_{\mathrm{total}}-\left(E_{\mathrm{PET}}+E_{H_{2}O}\right)\right)\;$$
where E_Total_ represents the total energy of PET and H_2_O, E_PET_ represents the total energy of PET after removing H_2_O, and E_H2O_ represents the total energy of H_2_O after removing PET. The negative value of E_Interaction_ is E_Binding_. The larger the value of E_Binding_, the stronger the interaction. Table 3 shows the binding energies of PET/D5 and PET/H_2_O at 413 K.

The binding energy of PET and D5 is significantly larger than that of PET and H_2_O, which demonstrates that the interfacial interaction force between PET and D5 is greater than that between PET and H_2_O. This conclusion can also be obtained by observing their interfacial interaction models after MD simulation (Figure 9). As shown in Figure 9a, before annealing, H_2_O molecules are distributed around the PET molecular chains, and D5 molecules also surround the PET molecular chains. After annealing, very few H_2_O molecules can enter the PET molecular chain. In contrast, more D5 molecules (Figure 9b) can enter the PET molecular chain and they completely interact with each other.

### 3.6. Simulation Analysis of Solubility of PET Molecular Chains

Solubility parameters and cohesive energy density are important for evaluating the compatibility of different materials. Solubility parameter (δ), which is defined as the square root of the cohesive energy density (CED), has been widely used to predict the compatibility of polymer blends.
(7)$$\delta =\sqrt{\mathrm{CED}}$$

When the δ of the two substances is relatively close, that is, when Δδ is less than 2.05 (Cal^1/2^/cm^3/2^), the compatibility of the two substances is better. When Δδ is greater than 10.02 (Cal^1/2^/cm^3/2^), the two substances are incompatible [37,38].

In this simulation study, using the Forcite module in the Material studio simulation software and the CED function, the solubility parameter value of each component was simulated and calculated. Table 4 shows the simulated solubility parameters of PET, H_2_O, and D5. The MD results are in good agreement with the experimental values [39], suggesting the reliability of the MD simulation. PET and D5 have similar solubility parameters, indicating their good compatibility.

## 4. Conclusions

In this paper, MD simulation was used to study the changes in PET microstructure in H_2_O and D5. The study found that after adding H_2_O, the glass transition temperature of PET decreases, the diffusion coefficient increases, and the free volume increases. After adding D5, the glass transition temperature is further reduced, and it is easier to dye PET. Moreover, the diffusion coefficient and free volume are larger than that with water, indicating that the swelling effect of D5 on PET is more ideal than that of water, which can be explained by the results of the binding energy and the solubility parameters. The binding energy of PET/D5 is significantly larger than that of PET/H_2_O, which demonstrates that the interfacial interaction force between PET and D5 is greater than that between PET and H_2_O. PET and D5 have similar solubility parameters, indicating their good compatibility. This study of the microstructure of PET is helpful to deepen the understanding of the mechanism involved during the process of dyeing PET with water and non-aqueous solvents. It can provide strong support for further research on dyeing PET with non-aqueous solvents.

## Figures and Tables

**Figure 1 materials-15-02148-f001:**
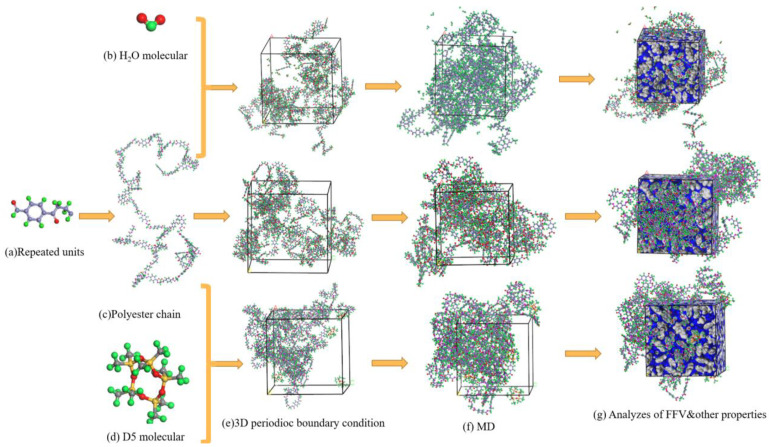
Models for molecular simulation (gray atom is C, green atom is H, and red atom is O).

**Figure 2 materials-15-02148-f002:**
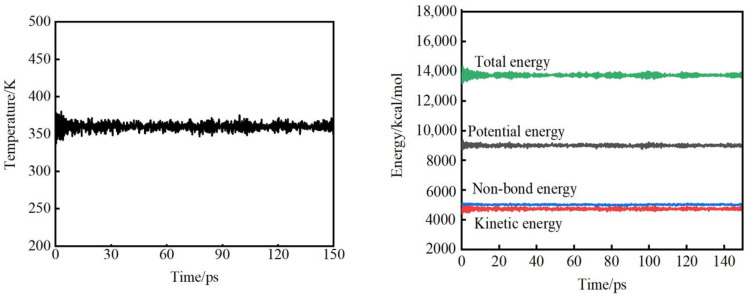
Temperature and energy curves of PET system at 360K.

**Figure 3 materials-15-02148-f003:**
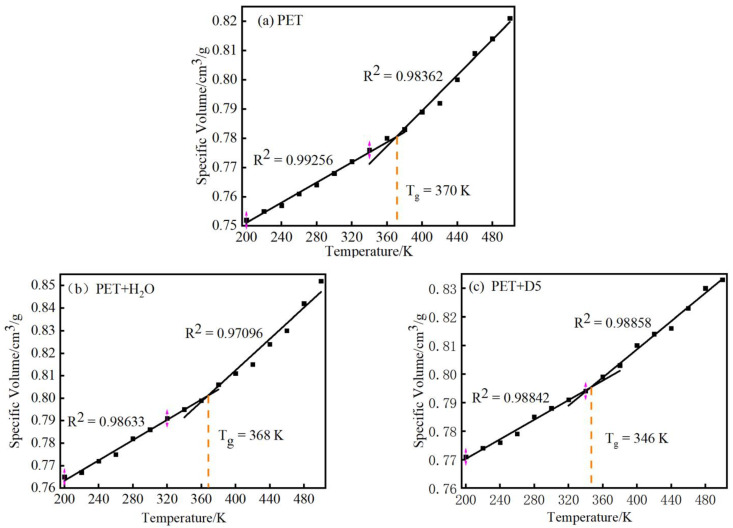
Specific Volume—Temperature MD curve.

**Figure 4 materials-15-02148-f004:**
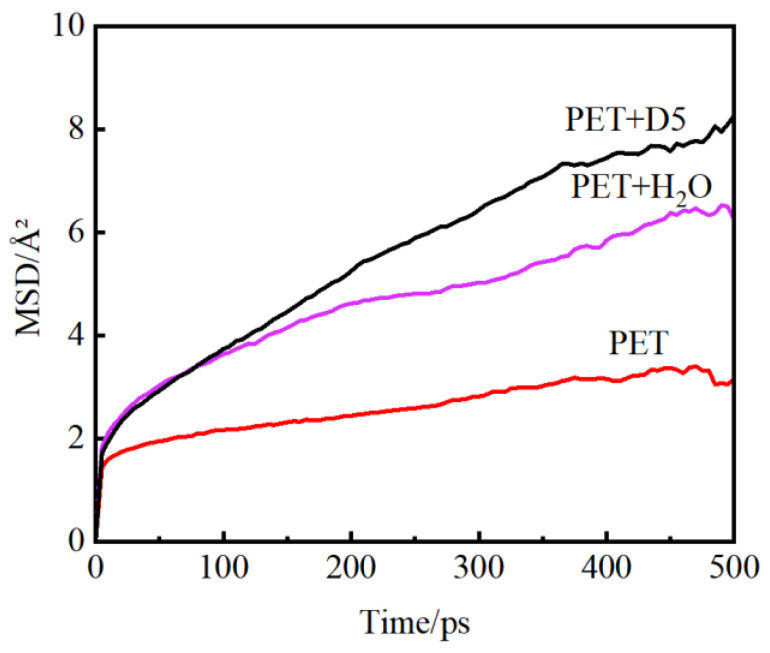
Mean square displacement curves of macromolecular chains in PET and swollen PET at 413 K.

**Figure 5 materials-15-02148-f005:**
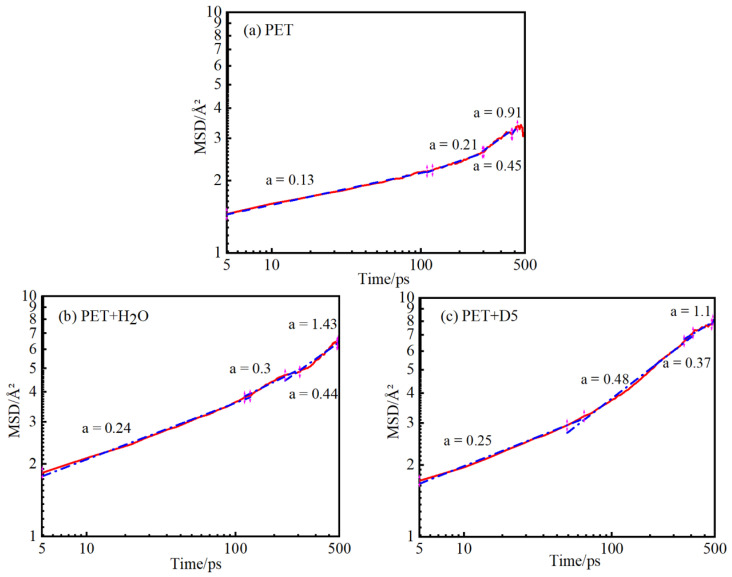
Segmental linear fitting diagrams of the Log10 curves of the MSD at 413 K for the three systems.

**Figure 6 materials-15-02148-f006:**
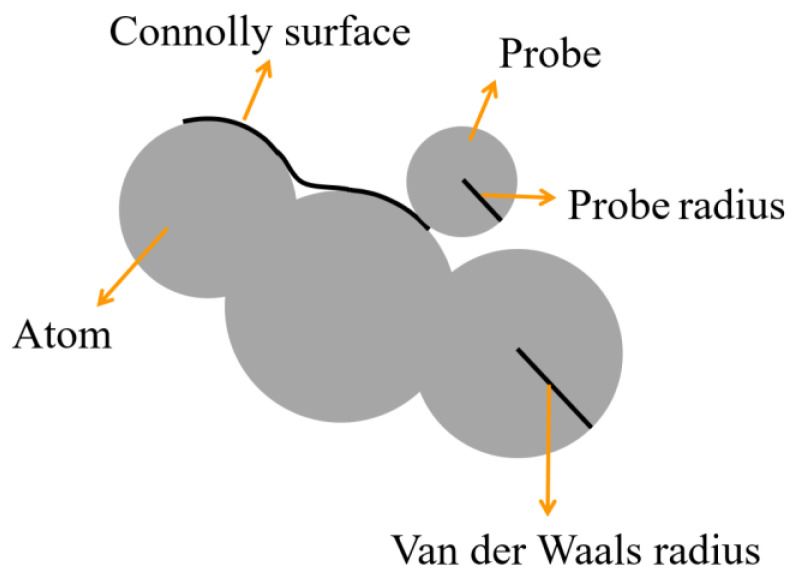
Schematic of van der Waals surface and Connolly surface.

**Figure 7 materials-15-02148-f007:**
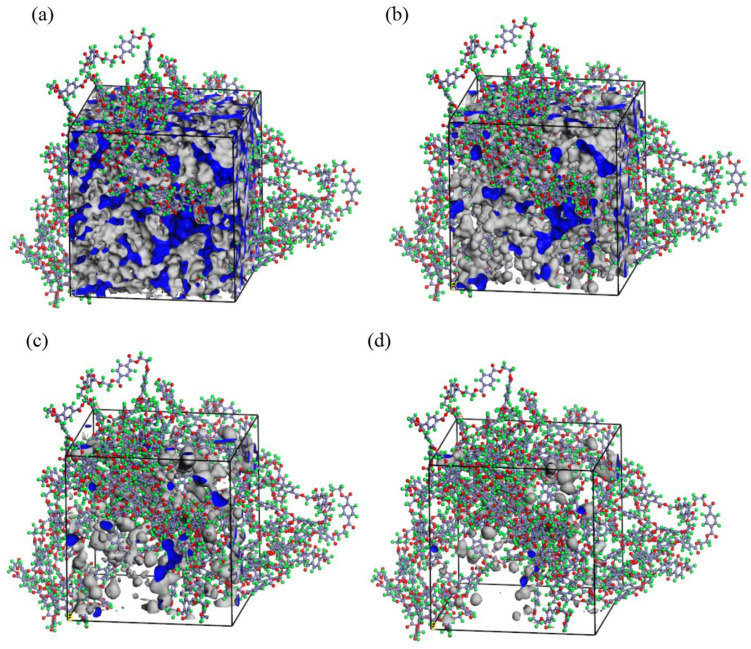
Free volume molecular model diagram of PET molecular chain at 413 K ((**a**) d = 1 Å; (**b**)d = 2 Å; (**c**) d = 3 Å; (**d**) d = 4 Å).

**Figure 8 materials-15-02148-f008:**
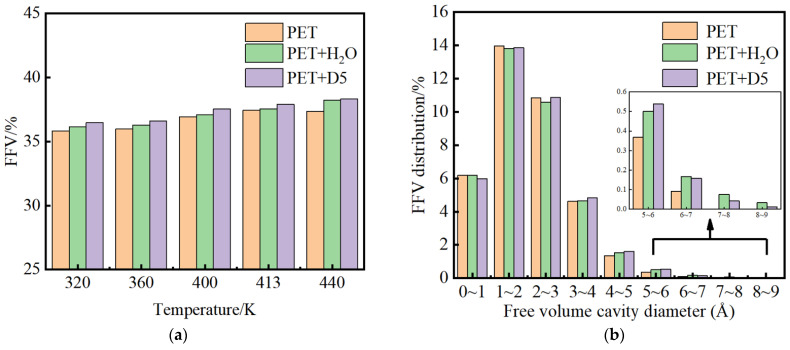
(**a**). Free volume fraction of PET in three systems (**b**). FFV distribution in three systems at 413 K at different temperatures.

**Figure 9 materials-15-02148-f009:**
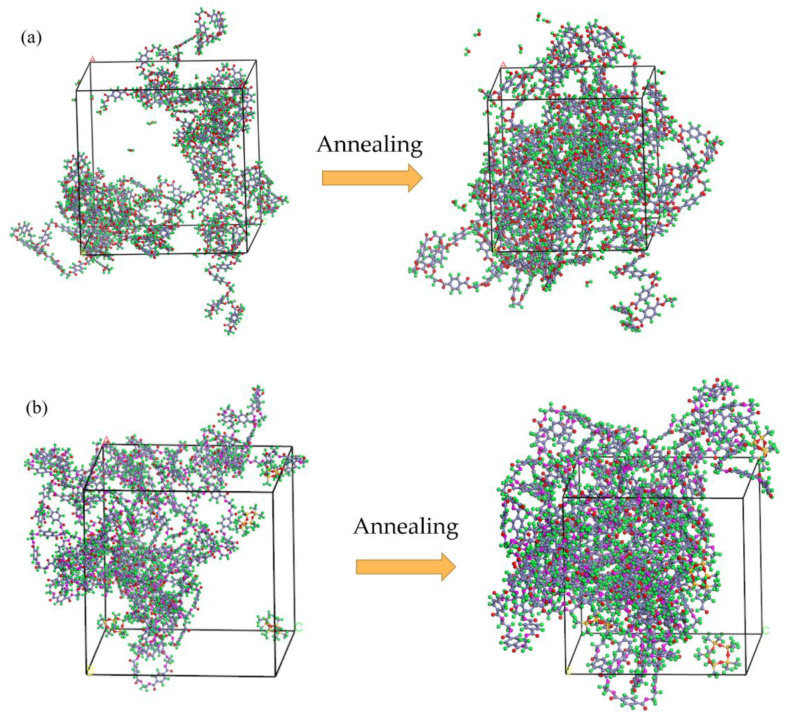
Comparison of solvent molecular positions before and after annealing ((**a**) PET/H_2_O and (**b**) PET/D5).

**Table 1 materials-15-02148-t001:** Basic parameters for MD of PET and its combined system with H_2_O and D5 solvent.

System	Cell Component	Weight Fraction of Components (w/%) Cell after Refinement				
		PET	D5/H_2_O	Density (ρ/10^3^ kg·m^−3^)	Volume (V/Å^3^)	Cell Size/Å^3^
1	4 PET chains	100	0	1.241	50,839.980	36.71^3^
2	4 PET chains, 60 H_2_O	97.2	2.8	1.230	53,345.383	37.09^3^
3	4 PET chains, 4 D5	96.3	3.7	1.239	53,495.574	37.28^3^

**Table 2 materials-15-02148-t002:** Diffusion coefficients of the PET backbone in PET and swollen PET.

Temperature (T/K)	Diffusion Coefficients of PET Main Chains in Three Systems (D/10^−8^ cm^2^ s^−1^)		
	PET	PET + H_2_O	PET + D5
320	0.98 × 10^−3^	0.94 × 10^−3^	0.99 × 10^−3^
360	0.99 × 10^−3^	1.07 × 10^−3^	1.79 × 10^−3^
400	1.15 × 10^−3^	1.16 × 10^−3^	2.36 × 10^−3^
413	1.22 × 10^−3^	1.37 × 10^−3^	2.09 × 10^−3^
440	1.23 × 10^−3^	1.75 × 10^−3^	2.56 × 10^−3^

**Table 3 materials-15-02148-t003:** The binding energies of PET/H_2_O and PET/D5 at 413 K (kcal/mol).

System	Energy (kcal/mol)					
	E_total_	E_polymer_	E_H2O_	E_D5_	E_interaction_	E_binding_
PET/H_2_O	9450.79	9774.79	−366.21	0	42.21	−42.21
PET/D5	8168.45	9774.79	0	−1567.567	−38.77	38.77

**Table 4 materials-15-02148-t004:** Solubility parameters of each system.

Components	Solubility Parameter (*δ*/Cal^1/2^/cm^3/2^)	
	Simulated Results *δ*_MD_	Experimental Results *δ*_Exp_
PET	19.5	19.9
H_2_O	50.7	47.8
D5	12.17	12.42

## Data Availability

Data is contained within the article.

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
