# Peer review of "Molecular Dynamics Simulation and Structure Changes of Polyester in Water and Non-Aqueous Solvents"

_materials, 2022, doi:10.3390/ma15062148_

Round 1

Reviewer 1 Report

In general manuscript is well written and it seems sound from scientific point of view. There are couple of things needed to be explained:

1) English language must be revised. For example:

a) Line 12. “…which can help TO reduce water…”

b) Line31. There to many infinitive forms are used throughout the manuscript like to+verb. In some places, instead of “to solve”, better to use “in order to solve…” or other forms.

c) Revise the statements at lines 49-51.

d) Line 65. “…are formed…” must be corrected as “…were formed…”.

e) Live a space between paragraphs.

f) Figure1, check the statements “repeat units or repeated unites”, “Analysics or analyzes or analytics”.

g) Line 36, “Rationality or rationality”. Check it please.

h) Line 164, is or are? Check it.

Check the rest of the manuscript in the similar way.

2) In Table 2, there are two systems, one has a positive binding energy, the other is negative. What does that mean? Signs say what? For example, what kind of interaction, exothermic or endothermic????

3) Please check the reference style. Some names are written with uppercase letters, while some lowercase.

4) Conclusion section should be enhanced.

Author Response

Dear Reviewers:

Thank you for your letter and for the reviewers’ comments concerning our manuscript entitled “Molecular dynamics simulation and structure changes of poly-ester in water and non-aqueous solvents” (Manuscript ID: materials-1572525). Those comments are all valuable and very helpful for revising and improving our paper, as well as the important guiding significance to our researches. We have studied comments carefully and have made correction which we hope meet with approval. Revised portion are marked in red in the paper. The main corrections in the paper and the responds to the reviewer’s comments are as flowing:

Reviewer 1:

Comments and Suggestions for Authors

In general manuscript is well written and it seems sound from scientific point of view. There are couple of things needed to be explained:

1) English language must be revised. For example:

  1. a) Line 12. “…which can help TO reduce water…”
  2. b) Line31. There to many infinitive forms are used throughout the manuscript like to+verb. In some places, instead of “to solve”, better to use “in order to solve…” or other forms.
  3. c) Revise the statements at lines 49-51.
  4. d) Line 65. “…are formed…” must be corrected as “…were formed…”.
  5. e) Live a space between paragraphs.
  6. f) Figure1, check the statements “repeat units or repeated unites”, “Analysics or analyzes or analytics”.
  7. g) Line 36, “Rationality or rationality”. Check it please.
  8. h) Line 164, is or are? Check it.

Check the rest of the manuscript in the similar way.

Response 1: Thank you for the reviewer’s comments. We are very sorry for our grammar mistakes. Some changes have been listed below:

  1. a) Line 12, the statements of “…which can help reduce water…” were corrected as “…which can help to reduce water…”
  2. b) Line 31, the statements of “to solve” were corrected as “in order to solve…”.
  3. c) lines 49-51have been revised
  4. d) “…are formed…” has been changed to “…were formed…”
  5. e) Blank lines have been added between paragraphs
  6. f) It has been modified in Figure 1
  7. g) We are very sorry for our silly error. “Rationality” has been changed to “rationality”
  8. h) We are very sorry for our silly error. “is” has been changed to “are”

Considering the space and precise, please see the vevised manuscrip for all the changes. Thanks!

2) In Table 2, there are two systems, one has a positive binding energy, the other is negative. What does that mean? Signs say what? For example, what kind of interaction, exothermic or endothermic????

Response 2: Thank you for the comments. The negative binding energy in Table 2 indicates that energy is required for the combination of PET and water, and the positive binding energy indicates an exothermic reaction, which can be combined without energy.

3) Please check the reference style. Some names are written with uppercase letters, while some lowercase.

Response 3: Thank you for the reviewer’s comments. We are very sorry for our incorrect writing, we have double checked the references and corrected capitalization errors. The exact location of the revisions has been noted in the revised manuscript.

4) Conclusion section should be enhanced.

Response 4: Thank you for your kind advice and meticulous approach. We have cut and rearranged the conclusion section, and made significant revisions. The exact location of the revisions has been noted in the revised manuscript.

 We tried our best to improve the manuscript and made some changes in the manuscript.  These changes will not influence the content and framework of the paper. And here we did not list the changes but marked in red in revised paper.
We appreciate for Reviewers’warm work earnestly, and hope that the correction will meet with approval.
Once again, thank you very much for your comments and suggestions

Reviewer 2 Report

The manuscript of Zheng an coworkers presents a computational study of the interaction between a small unit of polyethylene terephtalate (PET) with water and decamethylcyclopentasiloxane (D5) to investigate the influence of solvent on the swelling properties of the material. To achieve this goal a number of classical molecular dynamics simulations have been carried out with special focus on the glass transition temperature, solvent diffusion, the interaction energy, solubility and free volume analysis.

The work represents a original contribution that is well conducted. The preparation of figures and tables as well as the cited reference appear adequate for the most part. The employed methodology has been well-described.

Nevertheless, the manuscript would benefit from a number of improvements, that should be addressed in a minor revision:

1) The level of English throughout the manuscript is highly adequate. However, in parts the inclusion of articles (a, the) seems appropriate. Similarly, some phrases suffer from a wrong use of singular/plural (e.g. was vs were). The authors are asked to improve the text body in this regard.

2) The authors should check whether the abbreviation PE vs PET are adequately used throughout the manuscript (see for example the abstract).

3) The description of the methods is in the most parts adequate, however, the entire section 2 contains a really small number of references. Critical methods such as the COMPASS force field, the Anderson and Berendsen methods ,etc. should be adequately cited.

In the same section the definition of NPT should be on the first occurrence in the text (presently it is at the second). The abbreviation FFV should be explained.

4) In addition to the citations added under point 3 above it would be helpful for the readers to provide more information in the methods section about the COMPASS force field. Similar, the description of the analysis methods such as the diffusion coefficients, the free volume analysis, the calculation of the binding energy etc. is better placed in the methods section, so the results can indeed focus on the discussion of results. A better explanation how the cohesive energy density is determined should be included as well.

5) Figure 8 and 9: It would be a good idea to combine these figues into Figure 8 a and b. There is no need to present this data as two separate figures. Also it appears that there is a typo "caity" in the x-axis label of Figure 9.

Author Response

Dear Reviewers:

Thank you for your letter and for the reviewers’ comments concerning our manuscript entitled “Molecular dynamics simulation and structure changes of poly-ester in water and non-aqueous solvents” (Manuscript ID: materials-1572525). Those comments are all valuable and very helpful for revising and improving our paper, as well as the important guiding significance to our researches. We have studied comments carefully and have made correction which we hope meet with approval. Revised portion are marked in red in the paper. The main corrections in the paper and the responds to the reviewer’s comments are as flowing:

Reviewer 2:

Comments and Suggestions for Authors

The manuscript of Zheng an coworkers presents a computational study of the interaction between a small unit of polyethylene terephtalate (PET) with water and decamethylcyclopentasiloxane (D5) to investigate the influence of solvent on the swelling properties of the material. To achieve this goal a number of classical molecular dynamics simulations have been carried out with special focus on the glass transition temperature, solvent diffusion, the interaction energy, solubility and free volume analysis.

The work represents a original contribution that is well conducted. The preparation of figures and tables as well as the cited reference appear adequate for the most part. The employed methodology has been well-described.

Nevertheless, the manuscript would benefit from a number of improvements, that should be addressed in a minor revision:

1)The level of English throughout the manuscript is highly adequate. However, in parts the inclusion of articles (a, the) seems appropriate. Similarly, some phrases suffer from a wrong use of singular/plural (e.g. was vs were). The authors are asked to improve the text body in this regard.

Response 1: Thank you for the reviewer’s comments. We are very sorry for our negligence of the singular and plural in articles and phrases, The exact location of the revisions has been noted in the revised manuscript.

2) The authors should check whether the abbreviation PE vs PET are adequately used throughout the manuscript (see for example the abstract).

Response 2: Thank you for the reviewer’s comments. We are very sorry for our incorrect writing, the abbreviations in the article have been checked and revised.

3) The description of the methods is in the most parts adequate, however, the entire section 2 contains a really small number of references. Critical methods such as the COMPASS force field, the Anderson and Berendsen methods ,etc. should be adequately cited.

In the same section the definition of NPT should be on the first occurrence in the text (presently it is at the second). The abbreviation FFV should be explained.

Response 3: Thank you for the reviewer’s comments. Considering the Reviewers suggestion, some references about the COMPASS force field, the Anderson and Berendsen methods have been added to the article; after inspection, it is found that the positions of NPT and FFV explanations are indeed wrong, and they have been modified in the text.

4) In addition to the citations added under point 3 above it would be helpful for the readers to provide more information in the methods section about the COMPASS force field. Similar, the description of the analysis methods such as the diffusion coefficients, the free volume analysis, the calculation of the binding energy etc. is better placed in the methods section, so the results can indeed focus on the discussion of results. A better explanation how the cohesive energy density is determined should be included as well.

Response 4: Thank you for the reviewer’s comments. We give a detailed introduction to the COMPASS force field in the method section of part 2 on page 2 of the article, as in the text.

In the COMPASS force field, the total energy of the system is expressed as

 Etotal=Eb+Eθ+Eφ+ Eχ+Ecross+Eele+Evdw

Where Etotal is the total energy, Eb is bond stretching energy, Eθ is angle bending energy, Eφ is dihedral torsion energy, Eχ is out-of-plane energy, Ecross is crossterm interaction energy, Eele is electrostatic interaction energy, Evdw is van der Waals interaction energy.

We have read a lot of literature, and we also follow the discussion part about the description of the analysis methods such as the diffusion coefficients, the free volume analysis, the calculation of the binding energy etc. so that readers can understand the discussion of this article more clearly[1,2]

[1] Xw A , Meng S B , Sl C , et al. Analysis of phthalate plasticizer migration from PVDC packaging materials to food simulants using molecular dynamics simulations and artificial neural network[J]. Food Chemistry, 317.

[2] Yin C , Zhao X , Zhu J , et al. Experimental and molecular dynamics simulation study on the damping mechanism of C5 petroleum resin/chlorinated butyl rubber composites[J]. Journal of Materials ence, 2019.

We cover the cohesive energy density in more detail in the article:

When Δδ is less than 2.05(Cal1/2/cm3/2), the compatibility of the two substances is better. When Δδ is greater than 10.02(Cal1/2/cm3/2), the two are incompatible.

5) Figure 8 and 9: It would be a good idea to combine these figues into Figure 8 a and b. There is no need to present this data as two separate figures. Also it appears that there is a typo "caity" in the x-axis label of Figure 9.

Response 5: Thank you for the reviewer’s comments. We have made correction according to the Reviewer’s comments, the pictures and spelling in the article have been revised and marked.

We tried our best to improve the manuscript and made some changes in the manuscript.  These changes will not influence the content and framework of the paper. And here we did not list the changes but marked in red in revised paper.
We appreciate for Reviewers’warm work earnestly, and hope that the correction will meet with approval.
Once again, thank you very much for your comments and suggestions

Reviewer 3 Report

he authors performed the molecular dynamics simulation for polyester (PET) in solvents by using Material Studio 8.0 software. They calculated some quantities characterizing microscopic structure of polyester in solvents to study swelling mechanism.

The results shown in this manuscript are interesting.  However, I have the following requests.

  1. Introduction of this manuscript insists that the aim of the calculation in this manuscript is to improve PET dyeing. However, the relationship between the calculated results and dyeing is not shown.  The relationship should be shown in this manuscript.
  2. The glass transition temperature is obtained from the intersection of the two straight lines of the specific volume versus temperature plot in Section 3.2. Why is the temperature at the intersection glass transition temperature?  The reason or references mentioning the reason should be shown.
  3. At the grass transition temperature, the diffusion coefficient is expected to change drastically. The change of the diffusion constant is a clear evidence of the grass transition.  The grass transition should be discussed on the basis of the "dynamical" quantities such as the diffusion coefficient.   At least, the relationship between the grass transition temperature discussed in Section 3.2 and the diffusion coefficient should be mentioned.

Author Response

Dear Reviewers:

Thank you for your letter and for the reviewers’ comments concerning our manuscript entitled “Molecular dynamics simulation and structure changes of poly-ester in water and non-aqueous solvents” (Manuscript ID: materials-1572525). Those comments are all valuable and very helpful for revising and improving our paper, as well as the important guiding significance to our researches. We have studied comments carefully and have made correction which we hope meet with approval. Revised portion are marked in red in the paper. The main corrections in the paper and the responds to the reviewer’s comments are as flowing:

Reviewer 3:

Comments and Suggestions for Authors

The article by Jin Zheng, Dongshuang Wang, Qi Zhang, Meng Song, Mingli Jiao, Zhicheng Zhang is devoted to investigating the swelling mechanism of polyester in water and non-aqueous solvents. All investigations were done using molecular dynamics. Unfortunately, the current edition of the article contains many significant shortcomings, which does not allow me to recommend it for further discussion.

The article should be revised and submitted again. My comments are below.

1)Line 24, the authors write, “The cross-section of the PET molecule is circular under the microscope.” Perhaps a mistake was made here, and perhaps the authors did not mean molecules but fibers.

Response 1: Thank you for your suggestion. After inspection, it is found that it should be PET fibers, not PET molecular. It has been modified and marked in the article.

2)Lines 31-32, the authors write, “To solve the above problems, it is necessary to increase the circulation of the medium and reduce the staining.” Why should “staining” be reduced?

Response 2: Thank you for the reviewer’s comments. We are very sorry for our negligence, Lines 31-32 has been modified and marked in the article.

3)Paragraph 49-73 should be rewritten to include examples of the use of molecular dynamics relevant to this study only.

Response 3: Thank you for the reviewer’s comments. Considering the Reviewer’s suggestion, lines 49-73 in the article have been revised, adding references to molecular dynamics simulations and marking them in the text.

4)Line 95, the authors write, “Finally, the balanced cells.” Perhaps the authors meant equilibrated simulation boxes? In general, in the manuscript, it is recommended to use the generally accepted terms adopted in articles devoted to molecular dynamics.

Response 4: Thank you for the reviewer’s comments. We have made correction according to the Reviewer’s comments.

5) When describing the protocol for preparing material samples, their size should be indicated. The authors used relatively small simulation boxes. How will increasing the size of the system affect the results obtained?

Moreover, how many statistically independent systems at each parameter set were prepared to avoid the effect of the initial state on obtained results? It should be indicated.

Response 5: Thank you for the reviewer’s comments. We have added this

System

Cell component

Weight fraction of components (w/%)

Cell after refinement

PET

D5/H2O

Density

(ρ/103kg·m-3)

Volume(V/ Å3)

Cell Size/ Å3

1

4 PET chains

100

0

1.241

50839.980

36.713

2

4 PET chains, 60 H2O

97.2

2.8

1.230

53345.383

37.093

3

4 PET chains,4 D5

96.3

3.7

1.239

53495.574

37.283

part according to the Reviewer’s suggestion.

Table 1. Basic parameters for MD of PET and its combined system with H2O and D5 solvent

Three sets of parallel experiments were performed for each model, and we will consider the impact of this issue in more depth in the following studies.

6) In Table 1 and the text, it is better to use generally accepted units of measurement used in molecular dynamics (angstroms, etc.).

Response 6: Thank you for the reviewer’s comments. As suggested by the reviewers, the units in Table 1 have been revised.

7) Lines 109-112, the authors write, “There are two main criteria for judging whether the system reaches equilibrium: one is temperature equilibrium, which requires that the fluctuation of the equilibrium value should not exceed 10K. The second is energy balance, which requires the energy to be constant or fluctuate up and down along a constant value.”

It should be noted that the criteria used by the authors are necessary but not sufficient conditions. Since the goal of the work was to study the microporous structure, the criterion could be, for example, the achievement of an equilibrium of the conformational properties of molecules (the most accurate way) or pore sizes.

Response 7: Thank you for the reviewer’s comments. This temperature and energy are judged in most simulations, e.g. cited in the literature

 [1] Yang D , Zhao X , Chan T , et al. Investigation of the damping properties of hindered phenol AO-80/polyacrylate hybrids using molecular dynamics simulations in combination with experimental methods[J]. Journal of Materials Science, 2016.

Thanks for the suggestion, we can next try to judge the balance by pore sizes.

8) Lines 109-112, the authors write, “This also verified the rationality of the proposed model.” Did authors mean the correctness of the proposed model?

Response 8: Thank you for the reviewer’s comments. This also verifies the correctness of the model.

9) According to Figure 5 (it should be rebuilt on a logarithmic scale in all axes), the mean square displacement of macromolecular chains did not reach the diffusion mode, which appears as <dr^2(dt)> ~ dt. Obviously, the authors used insufficient time for modeling the systems under study. Therefore, estimates of the diffusion coefficient are questionable.

Response 9: Thank you for the reviewer’s comments. due to the impact of the epidemic, it is temporarily impossible to continue the test, but according to Figure 3, the model has become balanced, and we will extend the simulation time in the future.

10) It is not entirely clear what the authors mean in the caption to Figure 10? The caption to the figure should be made more meaningful in meaning. Also, the description of the result obtained in the text of the article should be made more explicit.

Response 10: Thank you for your reminder. Figure 10 (now Figure 9) in the text shows the changes in the molecular chain structures of PET/H2O and PET/D5 before and after the molecular dynamics simulation. It can be easily seen from the model that the interface interaction between D5 and PET is greater than water.

11) Describing data results in subsections:

 3.1. Evaluation of system balance (equilibration?);

3.2 Determination of the glass transition temperature of PET;

3.3 Mean Square Displacement;

3.4 Free Volume;

3.5 Binding energy;

3.6 Solubility parameters;

should be rewritten.

At the beginning of section 3, the motivation should be clearly stated. Why were these characteristics calculated? At the end of each subsection, the authors should discuss what the obtained results prove (clarify) in the light of the problem set at the beginning of the article?

Response 11: Thank you for the reviewer’s comments. We have re-written this part according to the Reviewer’s suggestion

12) The conclusion should also be rewritten according to the results. The current edition of the article does not disclose the mechanism involved during dyeing PET with water and non-aqueous solvents.

Response 12: Thank you for the reviewer’s comments. We have made correction according to the Reviewer’s comments. the conclusion has been rewritten to add the mechanism involved during dyeing PET with water and non-aqueous solvents.

We tried our best to improve the manuscript and made some changes in the manuscript.  These changes will not influence the content and framework of the paper. And here we did not list the changes but marked in red in revised paper.
We appreciate for Reviewers’warm work earnestly, and hope that the correction will meet with approval.
Once again, thank you very much for your comments and suggestions

Reviewer 4 Report

The article by Jin Zheng, Dongshuang Wang, Qi Zhang, Meng Song, Mingli Jiao, Zhicheng Zhang is devoted to investigating the swelling mechanism of polyester in water and non-aqueous solvents. All investigations were done using molecular dynamics. Unfortunately, the current edition of the article contains many significant shortcomings, which does not allow me to recommend it for further discussion.

The article should be revised and submitted again. My comments are below.

1) Line 24, the authors write, “The cross-section of the PET molecule is circular under the microscope.” Perhaps a mistake was made here, and perhaps the authors did not mean molecules but fibers.

2) Lines 31-32, the authors write, “To solve the above problems, it is necessary to increase the circulation of the medium and reduce the staining.” Why should “staining” be reduced?

3) Paragraph 49-73 should be rewritten to include examples of the use of molecular dynamics relevant to this study only.

4) Line 95, the authors write, “Finally, the balanced cells.” Perhaps the authors meant equilibrated simulation boxes? In general, in the manuscript, it is recommended to use the generally accepted terms adopted in articles devoted to molecular dynamics.

5) When describing the protocol for preparing material samples, their size should be indicated. The authors used relatively small simulation boxes. How will increasing the size of the system affect the results obtained?

Moreover, how many statistically independent systems at each parameter set were prepared to avoid the effect of the initial state on obtained results? It should be indicated.

6) In Table 1 and the text, it is better to use generally accepted units of measurement used in molecular dynamics (angstroms, etc.).

7) Lines 109-112, the authors write, “There are two main criteria for judging whether the system reaches equilibrium: one is temperature equilibrium, which requires that the fluctuation of the equilibrium value should not exceed 10K. The second is energy balance, which requires the energy to be constant or fluctuate up and down along a constant value.”

It should be noted that the criteria used by the authors are necessary but not sufficient conditions. Since the goal of the work was to study the microporous structure, the criterion could be, for example, the achievement of an equilibrium of the conformational properties of molecules (the most accurate way) or pore sizes.

8) Lines 109-112, the authors write, “This also verified the rationality of the proposed model.” Did authors mean the correctness of the proposed model?

9) According to Figure 5 (it should be rebuilt on a logarithmic scale in all axes), the mean square displacement of macromolecular chains did not reach the diffusion mode, which appears as <dr^2(dt)> ~ dt. Obviously, the authors used insufficient time for modeling the systems under study. Therefore, estimates of the diffusion coefficient are questionable.

10) It is not entirely clear what the authors mean in the caption to Figure 10? The caption to the figure should be made more meaningful in meaning. Also, the description of the result obtained in the text of the article should be made more explicit.

11) Describing data results in subsections:

 3.1. Evaluation of system balance (equilibration?);

3.2 Determination of the glass transition temperature of PET;

3.3 Mean Square Displacement;

3.4 Free Volume;

3.5 Binding energy;

3.6 Solubility parameters;

should be rewritten.

At the beginning of section 3, the motivation should be clearly stated. Why were these characteristics calculated? At the end of each subsection, the authors should discuss what the obtained results prove (clarify) in the light of the problem set at the beginning of the article?

12) The conclusion should also be rewritten according to the results. The current edition of the article does not disclose the mechanism involved during dyeing PET with water and non-aqueous solvents.

Reviewer 5 Report

The paper by Jin Zheng et al. investigates the influence of water and D5 on polyester (PET) by means of molecular dynamics. The authors are able to calculate the solubility parameter, the glass transition and the behaviour of the free volume, in addition to other quantities, in the pure polymer and with addition of water and D5; they find that D5 shows better compatibility than water.

I have the following comments:

- line 134-138: concerning the glass transition (GT) temperatures: it should be useful to know, if possible, the uncertainties associated. In fact, on the one hand the GT temperature of the simulated polymer is found to be 370 K, which is considered ‘close’ to the experimentally measured value 342-350 K. On the other hand, a difference of 2 K with respect to the system PET+water is considered significant, since it is claimed that the TG temperature ‘decreases’. It seems that the criteria to assess a significant difference are quite elastic. PET + D5 shows a TG temperature of 346 K: is it possible to compare it with an experimental determination? This could be useful in order to validate the simulation.

- lines 146-147: on the basis of which criterion was a temperature of 413 K decided?

- lines 162-169: would it be possible to collect all the diffusion coefficients in a table?

- line 208: from figure 8 it seems that the free volume in PET remains stable from 400 K; why do the authors state that it increases?

- line 215-216: the sentence should be reformulated, it is hardly understandable

- Figure 4: the caption contains a typo: ‘curve’; Figure 9: ‘cavity’, instead of ‘caity’

- Figure 9: it should be useful that also the caption, not only the text, reported the temperature at which the distributions are calculated.

I suggest to publish the paper after the authors considered the above remarks.

Round 2

Reviewer 4 Report

The authors have corrected the article. However, it has several shortcomings that do not allow me to recommend it for publication.

Small remarks.

1) Line 32, the Authors write, "In order to solve the above problems, it is necessary to use other media with different  properties from water."

It is not clear where this conclusion comes from. What part water takes in the described process should be explained more clearly.

2) Line 177, the Authors write, "more energy, so the movement is more active.and at the same time, at high temperature, ".

There is an extra dot here.

3) Line 179, the Authors write, "In water and D5, the diffusion coefficient of D5 is higher than that of water."

Do the Authors mean the diffusion coefficient of PET?

More significant remarks.

4) My previous comment, "(5) When describing the protocol for preparing material samples, their size should be indicated. The authors used relatively small simulation boxes. How will increasing the size of the system affect the results obtained? Moreover, how many statistically independent systems at each parameter set were prepared to avoid the effect of the initial state on obtained results? It should be indicated."

Authors' response, "Three sets of parallel experiments were performed for each model, and we will consider the impact of this issue in more depth in the following studies."

If the Authors have carried out three series of calculations, this should be mentioned in the text of the article. Moreover, if several calculations have been performed with the same parameters, this should be used to estimate the calculation error for different results.

5) My previous comment," (9) According to Figure 5 (it should be rebuilt on a logarithmic scale in all axes), the mean square displacement of macromolecular chains did not reach the diffusion mode, which appears as <dr^2(dt)> ~ dt. Obviously, the authors used insufficient time for modeling the systems under study. Therefore, estimates of the diffusion coefficient are questionable."

Authors' response, "Response 9: Thank you for the reviewer's comments. due to the impact of the epidemic, it is temporarily impossible to continue the test, but according to Figure 3, the model has become balanced, and we will extend the simulation time in the future."

The achievement of equilibrium does not mean that the displacement of molecules has reached the normal diffusion behavior. Figure 5 evidences the lack of simulation time. From the data provided, in my opinion, the diffusion coefficients calculated by the Authors look unreliable. If the Authors cannot improve their results, they should be removed from the article. The Authors should check the results of other works where the diffusion coefficients were calculated accurately.

6) My previous comment, "(10) It is not entirely clear what the authors mean in the caption to Figure 10? The caption to the Figure should be made more meaningful in meaning. Also, the description of the result obtained in the text of the article should be made more explicit."

Authors' response, "Response 10: Thank you for your reminder. Figure 10 (now Figure 9) in the text shows the changes in the molecular chain structures of PET/H2O and PET/D5 before and after the molecular dynamics simulation. It can be easily seen from the model that the interface interaction between D5 and PET is greater than water."

It is not clear from the Figure what exactly has changed. The Authors should describe it in more detail in the Figure caption.

7) In conclusion, the authors write, "In this paper, MD is used to study the changes of PET microstructure in H2O and D5. The study found that: after adding H2O, the Tg of PET molecular chain decreases, the diffusion coefficient increases, and the free volume increases; After the addition of D5, the Tg is further reduced, the dyeing is easier, and the diffusion coefficient and free volume are larger than those of the water solvent. Increase the number of holes in PET from 2.0×10-10m to 5.0×10-10m. The solubility parameters, glass transition, free volume and other behaviors of adding water and D5 into PET found that the swelling effect of D5 for PET was more ideal than that of water solvent, and D5 showed better compatibility than water. The research on the microstructure of PET is beneficial to deepen the understanding of the mechanism in the dyeing process of water and non-aqueous solvents, and provide strong support for further research on non-aqueous solvent dyeing."

This conclusion is written very chaotically. It is not entirely clear what the authors studied, what problem they managed to solve, and how the results obtained confirm this. It should be rewritten completely.

At the same time, the Authors make the wrong statement, "the Tg of PET molecular chain decreases." Tg is the properties of the polymer system as a whole. The segmental mobility of the polymer chain (above and below Tg) changes.

Round 3

Reviewer 4 Report

The authors have corrected the article, but I have one significant comment.

My comment for the initial version of the article was, "(9) According to Figure 5 (it should be rebuilt on a logarithmic scale in all axes), the mean square displacement of macromolecular chains did not reach the diffusion mode, which appears as <dr^2(dt)> ~ dt. Obviously, the authors used insufficient time for modeling the systems under study. Therefore, estimates of the diffusion coefficient are questionable."

Authors have responded, "Response 9: Thank you for the reviewer's comments. due to the impact of the epidemic, it is temporarily impossible to continue the test, but according to Figure 3, the model has become balanced, and we will extend the simulation time in the future."

I did an explanation of my point on this response "The achievement of equilibrium does not mean that the displacement of molecules has reached the normal diffusion behavior. Figure 5 evidences the lack of simulation time. From the data provided, in my opinion, the diffusion coefficients calculated by the Authors look unreliable. If the Authors cannot improve their results, they should be removed from the article. The Authors should check the results of other works where the diffusion coefficients were calculated accurately."

The Authors reply again, "Response 5: Thank you for your comments. according to the energy and temperature curve seen in Figure 3, the simulation time was long enough. "

However, I cannot entirely agree with this answer because it contradicts calculating the diffusion coefficient. I have already advised the Authors to see the work of other authors where this was done correctly. Unfortunately, the Authors did not follow my advice. I made a selection of the articles I had read. The references placed below

DOI: 10.1103/PhysRevLett.109.178001 ,see FIG. 1b 

DOI: 10.1021/acs.jpcb.7b02502 , see Fig 9, 10

DOI: 10.1021/ma070201o , see Fig 8

DOI: 10.1021/ma0019508 ,see Fig.6

DOI: 10.3390/polym12040907 , see Fig 11. (open access)

Therefore, the authors have to improve their Fig. 5 and do it in a double-logarithmic plot of the displacement of the diffusant molecules r^2 vs. time computed (log(r^2) vs. log(t)) and show that the diffusive regime (where log(r^2) is roughly linear with log (time)) is eventually reached. If this is not satisfied, then the obtained results of the diffusion coefficient cannot be considered correct. So, if the results in Fig. 5 cannot be improved, then Fig. 5 may be removed from the article.
